# Weighted Mutual Learning with Diversity-Driven Model Compression

**Miao Zhang**[1,2]**, Li Wang**[5]*****, David Campos**[2]**, Wei Huang**[3]**, Chenjuan Guo**[4]**, Bin Yang**[4]
[1]Harbin Institute of Technology (Shenzhen)  [2]Aalborg University  [3]UNSW
[4]East China Normal University  [5]Shandong First Medical University
zhangmiao@hit.edu.cn; dgcc@cs.aau.dk; weihuang.uts@gmail.com
{cjguo, byang}@dase.ecnu.edu.cn; li.wang-9@student.uts.edu.au

## Abstract

Online distillation attracts attention from the community as it simplifies the traditional two-stage knowledge distillation process into a single stage. Online distillation collaboratively trains a group of peer models, which are treated as students, and all students gain extra knowledge from each other. However, memory consumption and diversity among students are two key challenges to the scalability and quality of online distillation. To address the two challenges, this paper presents a framework called Weighted Mutual Learning with Diversity-Driven Model Compression (**WML**) for online distillation. First, at the base of a hierarchical structure where students share different parts, we leverage the structured network pruning to generate diversified students with different models sizes, thus also helping reduce the memory requirements. Second, rather than taking the average of students, this paper, for the first time, leverages a bi-level formulation to estimate the relative importance of students with a close-form, to further boost the effectiveness of the distillation from each other. Extensive experiments show the generalization of the proposed framework, which outperforms existing online distillation methods on a variety of deep neural networks. More interesting, as a byproduct, **WML** produces a series of students with different model sizes in a single run, which also achieves competitive results compared with existing channel pruning methods.

## 1  Introduction

Deep neural networks (DNNs) have achieved impressive success for a variety of application fields, including computer vision [27, 46], natural language processing [2], time-series analytics [53], and so on [6, 48, 54]. However, the performance of DNNs is heavily dependent on model parameters and computations, which hinders its application on intelligent edge systems [20]. To relieve this issue, various techniques have been proposed to design lightweight models, including network pruning [18], network quantization [7], and neural architecture search [43]. Knowledge distillation [25, 33, 51], as a complementary method to further improve the performance of the lightweight models, can transfer the discriminative ability from a high-capacity yet cumbersome *teacher* model to a small and compact *student* model by encouraging the student to mimic the behavior of the teacher. The vanilla knowledge distillation [25], as shown in Figure 1 (a), includes two stages: it first trains the cumbersome teacher model, whose soft logits that contain informative dark knowledge [25] are then transferred to a compact student model. As the teacher is fixed during the knowledge transferring, the student can only learn from the teacher once it is trained, and the vanilla knowledge distillation is also called offline distillation [15]. To further simplify the distillation process and relieve the restriction when the larger-capacity teacher model is not available, the online distillation framework has been

---

*Corresponding Author.

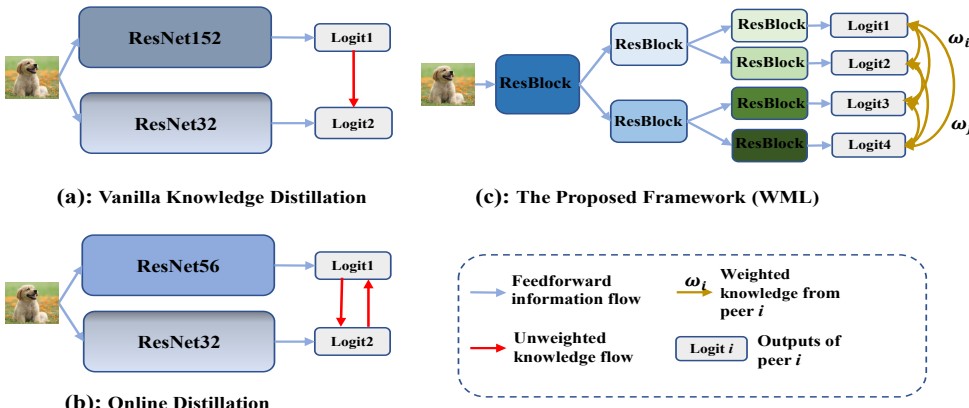

Figure 1: **Knowledge distillation, online distillation, and the proposed WML**. (a) [25] is the first knowledge distillation, where the student ResNet32 can only learn from the trained teacher ResNet152. (b) [70] is also called deep mutual learning, where the students can learn from each other during the training. (c) is the proposed **WML** framework with a hierarchy structure, where different colored Resblocks are pruned under different ratios to enable diversified peer models and also reduce the memory consumption. Rather than averaging the knowledge from peers, **WML** leverages bi-level optimization to estimate the relative importance of each peer (e.g., weight $\omega_j$ for peer $j$) in forming a more informative knowledge source.

proposed [4, 70, 71], which simultaneously trains a group of student models (a.k.a. peers) with similar structures. As each student can gain extra knowledge from each other, it is also called deep mutual learning [70], as shown in Figure 1 (b). Although online distillation conducts the distillation with a one-phase training scheme without pretraining a teacher, it has high memory consumption when jointly training several independent models. In addition, it is supposed to yield more informative knowledge when diversity presents among students, while the similar structures in online distillation may hinder the students from learning more informative knowledge from peers [9, 16].

In order to simultaneously overcome the two aforementioned difficulties, memory consumption and diversity among peers, this paper proposes a novel framework, called Weighted Mutual Learning with Diversity-Driven Model Compression (**WML**) for online distillation, as present in Figure 1 (c). Our WML considers a hierarchical structure where each peer share different parts, so as reduce the memory consumption compared with online distillation. In addition, we leverage the structured network pruning to generate diversified peer models under different pruning ratios, which also further reduces the memory consumption. On the other hand, simply treating the peers with different capacities to be equally important might neglect the diversity of all peer models, and limit the effectiveness of online distillation when learning from a group of peers with different capacities [9, 16]. In this paper, we formulate the weighted mutual learning based online distillation as a bilevel optimization problem, where the inner loop is to learn compact peers with the online distillation and the outer loop is to find suitable weights to determine a better optimization direction for knowledge distillation from a group of peers. The weights are optimized by the mirror descent [3, 5], and the hypergradient is analyzed in a theoretical way with a close-form. Our formulation can be seen as a dynamic weighting strategy for online distillation, which adaptively distills knowledge from the peer models to a student. A summary of our main contributions follows.

- First, this paper proposes a Weighted Mutual Learning with Diversity-Driven Model Compression (**WML**) for online distillation, where the peer models share different parts in a hierarchical structure and the structured network pruning is leveraged to further generate diversified peers and reduce the memory consumption.

- Second, we formulate the weighted mutual learning based online distillation as a bilevel optimization problem, and the hypergradient of optimizing the weights is derived with a close-form. The dynamic weighting strategy enables the online distillation to adaptively determine the importance of each peer in distilling knowledge from a group of peers.

- Third, extensive experiments on several neural networks and three datasets verify the effectiveness of the proposed framework, which significantly outperforms existing online distillation and self-distillation methods. More interesting, as a byproduct, **WML** can get a series of pruned student models with different model sizes in a single run, which also achieves competitive results when compared with existing channel pruning methods.

## 2 Related work

### 2.1 Knowledge Distillation

Knowledge Distillation (KD) is the concept [25] that distills knowledge from a larger *teacher* model to help the training of a smaller *student* model, by steering the student's logits towards teacher's logits, as shown in Figure 1 (a). FitNet [44] further extends the feature maps as the learning target instead of logits. Rather than only learning from a single teacher, many recent studies [9, 16] investigate learning from a group of teachers, where the outputs of teachers are usually averaged to guide the student training, while how to effectively integrate different types of knowledge from multiple teachers is less investigated [9, 16]. The traditional knowledge distillation is also called offline distillation since the teacher models are first trained before distillation. Differently, online distillation simultaneously trains a group of student models that can gain extra knowledge from each other without a pretrained teacher model, so it is also called the teacher-absent knowledge distillation. Deep mutual learning (DML) [70]is the first proposed online setting for knowledge distillation, which trains multiple peer models in a collaborative way that each model can be seen as a teacher during others' training process. KDCL [16] further ensembles the output of all students as the teacher knowledge in the online distillation setting since the ensemble is supposed to yield better generalization ability than a single model [1]. To further reduce the training requirements, several recent works [71] enforce all students to share the same early blocks while equipped with individual auxiliary head branches. Although this paradigm can relieve the heavy memory consumption in online distillation, it suffers from the drawback of diversity as all peers share the same backbone and with a similar structure. Differently, Song and Chai [47] proposed a hierarchical structure so that the peers can share different parts in the early blocks. Self-distillation is another recently-proposed distillation paradigm, which can be regarded as a special case of online distillation [15, 56, 61]. Specifically, [61] leverages the knowledge from the deeper layers to guide the training for the shallow sections in the backbone. In this paper, we consider a hierarchical structure [47] that each peer shares different parts and gains weighted knowledge from other peers instead of the average ensemble. Different from self-distillation that reduces the "depth" to get peers, this paper leverage the channel pruning to "slim" models to get diversified peers.

### 2.2 Network Pruning

Network pruning is an effective way to compress overparameterized neural networks by removing parameters with minimal performance degradation [57]. Based on the granularity, network pruning can be divided into two types: unstructured and structured pruning, where the former individually removes parameters in a network while the latter removes entire filters, channels, or even layers [40]. Although the unstructured pruning can mask most parameters in a network without performance drop [12], it requires specific hardware or software to accelerate the sparse network's inference time [10, 19]. Differently, structured pruning preserves the structural regularity and can accelerate network on commodity hardware platforms [22, 32]. In this paper, we consider structured pruning for generic applications. Next, the network pruning can happen after training vs. before training (a.k.a. at initialization) [49]. Most conventional pruning methods identify the redundant weights that are removed with least degrading the performance of the trained model. In contrast, recent works show that the randomly initialized neural network can be pruned without incurring any training [30], where SNIP [29], GraSP [50], and SynFlow [49] are three well-known metrics for the unstructured pruning at initialization. [40] further extends these metrics for structured pruning through summing scores of all parameters $\theta$ in a filter to determine its importance. In this paper, we extend SNIP to determine the pruning ratio for each layer at initialization as [40] demonstrates it is the most effective among the three to determine the filter importance for channel pruning at initialization.

# 3  Methodology

This section describes our Weighted Mutual Learning with Diversity-Driven Model Compression (**WML**) framework for online distillation, which consists of three main parts: the overall structure and the optimization for learning a group of peer models, the pruning approach to get a series of peers with different model sizes, and a bi-level formulation based weighted mutual learning.

## 3.1  Overview Structure and Loss Function

Figure 1 (c) depicts the overall structure of the proposed **WML** framework, where we take ResNet-32, which is divided into three sections in terms of depth, as an example for easy understanding. Different from the most commonly-used auxiliary heads [71] that all peer models share the same part, we consider a hierarchical structure [47] where each peer model shares different parts. The framework is similar to the binary tree, where the branches at the same levels are copies of each other, and each path is considered as a peer model. In addition, we further prune each block in **WML** with different ratios to further enhance the diversity among peer models and also reduce the memory consumption. Different from the vanilla knowledge distillation, which requires a much larger pretrained model as the teacher, the online distillation directly learns from each other or the ensembles of students [1]. In this paper, we consider the linear combination of knowledge from all peer models, and the weight of each peer model is determined by a bi-level formulation which will be discussed in Sec. 3.3

This paper focuses on the multi-class classification problems, and $X = \{x_n\}_{n=1}^N$ is the training images with $N$ samples, $Y = \{y_n\}_{n=1}^N$ is the corresponding labels. The output logit for $i$-th peer model is represented as $z_i$, and its corresponding cross entropy loss is $L_{CE}$ which is computed with the $z_i$ and labels $Y$. The overall loss to train the whole framework containing $M$ peers is defined as:

$$loss = (1-\alpha) \sum_{i=1}^M \omega_i \mathcal{L}_{CE}(z_i, Y) + \alpha \sum_{i=1}^M \sum_{j=1}^M \omega_j KL(z_i, z_j), \tag{1}$$

where $\omega_i$ indicates the importance of the $i$-th peer model and $\alpha$ is the hyperparameter to balance the supervision from labels and teachers. $\sum_{j=1}^M \omega_j KL(z_i, z_j)$ is the weighted knowledge from peers, which is one of the main differences between our WML and DML [70] in loss function, where DML considers each peer equally important in distillation. Another alternative distillation strategy is to calculate the KL divergence between each student with the ensemble of all students [16], while [70] found it is not as competitive as directly learning from multiple teachers.

## 3.2  Diversifying Peer Models with Channel Pruning

Although the hierarchical structure can proportionally reduce the memory consumption as peers share different blocks to reduce complexity, it still has higher computational cost than traditional knowledge distillation. To further reduce the memory requirements, we introduce the channel pruning, the most commonly-used paradigm in structured pruning, to our framework, which also encourage the diversity among homogenized peers. The training-free network pruning has shown its superiority in finding competitive subnetwork with extreme efficiency [29, 49, 50], and [40] has demonstrates SNIP [29] is informative to determine the filter importance for channel pruning. To be more specific, the relative importance of each filter $f$ based on SNIP can be calculated as: $\mathcal{I}_f = \sum_{\theta_i \in f} g(\theta_i) = \sum_{\theta_i \in f} \left| \frac{\partial \mathcal{L}(\theta_i)}{\partial \theta_i} \odot \theta_i \right|$, where $\left| \frac{\partial \mathcal{L}(\theta_i)}{\partial \theta_i} \odot \theta_i \right|$ is the SNIP score of parameter $\theta_i$. This value gives an expectation of the impact on loss when the filter is pruned, and filters with smaller SNIP scores tend to produce weak impact to the loss as compared to the other filters in that layer [40]. In this paper, we further propose a relative importance $\mathcal{I}_l$ to determine the importance of each layer at initialization, which is defined as:

$$\mathcal{I}_l = \frac{1}{N_l} \sum_{f \in l} \mathcal{I}_f, \tag{2}$$

where $N_l$ is the number of parameters in layer $l$. Given a pruning ratio $P$ of a model, the number of pruned parameters of each layer is then obtained by $p_l = P \times N \times \tilde{\mathcal{I}}_l$, where $\tilde{\mathcal{I}}_l$ is a normalization of $\frac{1}{\mathcal{I}_l}$ and $N$ is the total number of parameters in the model, and the number of preserved filter is

calculated accordingly which aims to preserve more filters in a more important layer. In the practical implementation, we set a upper bound of the pruned ratio for each layer to avoid layer collapse [50]. In this way, given different $P$ to peers, we can get a series of peer models with different structures.

### 3.3 Weighted Distillation with Bi-level Formulation

The proposed framework contains $M$ peer models whose outputs can be also combined to generate a powerful ensemble. However, the generalization capacity of the peer model is different from the model size. Intuitively, we would expect to raise the importance of model $i$ with the highest capacity in distillation, i.e., enlarging $\omega_i$ when peer model $i$ contains most parameters [4, 9]. In this paper, we consider a dynamic weighting method to determine the importance of each peer, and the weighted mutual learning for online distillation can be formulated as a bilevel optimization problem:

$$\min_{\omega} \ \mathcal{L}_{CE}(\sum_{i=1}^{M} \omega_i z_i^*, Y) \ \text{ s.t. } \theta^* = \text{argmin}_{\theta} \ (1-\alpha) \sum_{i=1}^{M} \omega_i \mathcal{L}_{CE}(z_i, Y) + \alpha \sum_{i=1}^{M} \sum_{j=1}^{M} \omega_j KL(z_i, z_j), \quad (3)$$

where $z_i^*$ is obtained by training the whole framework $\theta$ with loss function in Eq.(1). The inner loop is to optimize the whole network $\theta$ with the loss defined in Eq.(1), and the outer loop is to adjust the importance $\omega$ to make the weighted ensemble match the ground truth so that it reflects the importance of each peer more precisely. We define the loss function of inner loop optimization as $\mathcal{L}_1$ and outer loop as $\mathcal{L}_2$ for simplicity. Since $z_i^*$ is the local optimizer in the inner loop, Eq.(3) is a complicated nested optimization problem [37, 42, 54, 64, 68, 69]. In this paper, we consider a common one-step unroll learning paradigm [37, 42] that $\theta^* = \theta - \gamma \nabla_\theta \mathcal{L}_1$ with learning rate $\gamma$, and the gradient for the outer loop (which is also called as hypergradient) can be formulated as:

$$\nabla_\omega \mathcal{L}_2 = \frac{\partial \mathcal{L}_2}{\partial \omega} + \frac{\partial \mathcal{L}_2}{\partial \theta^*} \frac{\partial \theta^*(\omega)}{\partial \omega} = \frac{\partial \mathcal{L}_2}{\partial \omega} - \gamma \frac{\partial \mathcal{L}_2}{\partial \theta} \frac{\partial^2 \mathcal{L}_1}{\partial \omega \partial \theta}. \quad (4)$$

However, it is still impractical to calculate the second-order information for a large neural network, e.g., $\frac{\partial^2 \mathcal{L}_1}{\partial \omega \partial \theta}$. The finite-difference approximation is usually leveraged to roughly approximate this term [37], while it only considers the first two terms with the Taylor expansion. In the following theorem, we show that we can exactly calculate $\frac{\partial^2 \mathcal{L}_1}{\partial \omega \partial \theta}$ without approximation in our framework.

**Theorem 1** *With one-step unroll learning paradigm, the gradeint for $\omega_i$ in Eq.(3) is formulated as:*

$$g_{\omega_i} = \nabla_{\omega_i} \mathcal{L}_2 = \frac{\partial \mathcal{L}_2}{\partial \omega_i} - \gamma \frac{\partial \mathcal{L}_2}{\partial \theta} \frac{\partial^2 \mathcal{L}_1}{\partial \omega_i \partial \theta} = \frac{\partial \mathcal{L}_2}{\partial \omega_i} - \gamma \frac{\partial \mathcal{L}_2}{\partial \theta} \frac{\partial \mathcal{L}_a}{\partial \theta}^\mathsf{T}, \quad (5)$$

*where $\mathcal{L}_a = (1-\alpha)\mathcal{L}_{CE}(z_i, Y) + \alpha \sum_{j=1}^{M} KL(z_j, z_i)$.*

**Proof:** Based on the definition of $\mathcal{L}_1$ in Eq.(1), we have

$$\begin{aligned}
\frac{\partial \mathcal{L}_1}{\partial \omega_i} &= \frac{\partial((1-\alpha)\sum_{i=1}^{M} \omega_i \mathcal{L}_{CE}(z_i, Y) + \alpha \sum_{j=1}^{M} \sum_{i=1}^{M} w_i \cdot KL(z_j, z_i))}{\partial \omega_i} \\
&= (1-\alpha)\mathcal{L}_{CE}(z_i, Y) + \alpha \frac{\partial \sum_{j=1}^{M} \sum_{i=1}^{M} w_i \cdot KL(z_j, z_i)}{\partial \omega_i} \\
&= (1-\alpha)\mathcal{L}_{CE}(z_i, Y) + \alpha \sum_{j=1}^{M} KL(z_j, z_i).
\end{aligned} \quad (6)$$

Accordingly, $\frac{\partial^2 \mathcal{L}_1}{\partial \omega \partial \theta}$ can be calculate with *autograd* of $\mathcal{L}_a$ on $\theta$, and Theorem 1 is then proved. $\square$

Theorem 1 shows that the second-order term $\frac{\partial^2 \mathcal{L}_1}{\partial \omega \partial \theta}$ can be analyzed in a closed-form without Taylor expansion approximation when $\mathcal{L}_1$ is with a specific structure. Now that we can simply get the gradient to update model parameters $\theta$ and the weights for peers $\omega$ with *autograd*. Since $\omega$ is a probability simplex that $\sum_{i=1}^{M} \omega_i = 1$, we use the mirror descent to update $\omega$ [3, 5]. Algorithm 1 outlines our weighted mutual learning for online distillation. To be more specific, we first run several

**Algorithm 1** Weighted Mutual Learning (WML) for Online Distillation

1: **Input**: Dataset $\{(x_n, y_n)\}_{n=1}^N$; Given pruning ratios for each peer model $\{p_1, ..., p_M\}$.
2: Initialized hierarchical model $\theta^0$ and peer weights $\omega^0$;
3: Calculate the filter importance with SNIP at initialization, and prune peer models $i$ based on the given pruning ratios;
4: **for** $k = 1, ..., K$ **do**
5:     With the peer importance $\omega^k$, run $T$ steps of SGD to update the model parameters $\theta$ with the weighted loss function in Eq.(1);
6:     Calculate the gradient for $\omega^k$ based on Eq.(5);
7:     Run one step of mirror descent and update $\omega^k$ to get $\omega^{k+1}$ based on Eq.(7);
8: **end for**
9: **output:** $M$ models with outputs $\{z_1, ..., z_M\}$ the weights for peers $\omega$.

Table 1: Top-1 accuracy (%) comparison results with self-distillation on CIFAR100.

| Networks | Methods | Baseline | Model1 | Model2 | Model3 | Model4 | Ensemble |
|---|---|---|---|---|---|---|---|
| ResNet18 | DSN [28] | 77.09 | 67.23 | 73.80 | 77.75 | 78.38 | 79.67 |
| | SD [61] | 77.09 | 67.85 | 74.57 | 78.23 | 78.64 | 79.67 |
| | SCAN [62] | 77.09 | **71.84** | **77.74** | 78.62 | 79.13 | 80.46 |
| | **WML** | 77.09 | 71.15 | 75.65 | **78.88** | **79.38** | **80.56** |
| ResNet50 | DSN [28] | 77.68 | 67.87 | 73.80 | 74.54 | 80.27 | 80.67 |
| | SD [61] | 77.68 | 68.23 | 74.21 | 75.23 | 80.56 | 81.04 |
| | SCAN [62] | 77.68 | 73.69 | 78.34 | 80.39 | 80.45 | 81.78 |
| | **WML** | 77.68 | **78.58** | **79.69** | **80.81** | **81.24** | **82.57** |
| ResNet101 | DSN [28] | 77.98 | 68.17 | 75.43 | 80.98 | 81.01 | 81.72 |
| | SD [61] | 77.98 | 69.45 | 77.29 | **81.17** | 81.23 | 82.03 |
| | SCAN [62] | 77.98 | 72.26 | 79.26 | 80.95 | 81.12 | 82.06 |
| | **WML** | 77.98 | **79.60** | **81.16** | 81.14 | **81.46** | **83.03** |

steps of SGD based on the loss function in Eq.(1) to update model parameters $\theta$ with a fixed $\omega$. Then we calculate the gradient of $\omega_i$ based on Eq.5, and run one step of mirror descent to update $\omega_i$ that:

$$\omega_i^{k+1} = \frac{\omega_i^k \exp\left\{-\eta \nabla_{\omega_i^{k+1}} \mathcal{L}_2\right\}}{\sum_{i=1}^M \omega_i^k \exp\left\{-\eta \nabla_{\omega_i^{k+1}} \mathcal{L}_2\right\}}, \tag{7}$$

where $\eta$ is the step size, and $\omega_i^k$ is the model importance for *i-th* peer in *k-th* step. We repeat this loop until the stopping criterion meets.

## 4 Experiments

We perform a series of experiments to evaluate the effectiveness of our framework on seven convolutional neural networks with three image classification datasets. We first compare our framework with two self-distillation baselines, SD [61] and SCAN [62], and the deeply supervised net (DSN) [28], since they share similar structures as our WML. Specifically, SD and SCAN add early-exit blocks to the shallow blocks to get peer models, where SCAN contains additional attention modules. DSN shares a similar structure as SD while without knowledge distillation. After that, we compared WML with existing knowledge distillation and online distillation baselines. More interesting, as a byproduct, we also compared the pruned models produced by our framework with channel pruning baselines. A series of ablation studies on dynamic weighting strategy and pruning are also discussed.

### 4.1 Comparison with Self-Distillation

For fair comparisons in this experiment, we consider 4 peer models in our WML, and also prune peer models into similar model size as classifiers in self-distillation (SD) [61], where Model1-3 are under

Table 2: Top-1 accuracy (%) comparison results with self-distillation on ImageNet.

| Networks | Methods | Baseline | Model1 | Model2 | Model3 | Model4 | Ensemble |
|---|---|---|---|---|---|---|---|
| MobileNetV2 | SD [61] | 71.52 | 52.23 | 61.33 | 68.64 | 72.37 | 72.73 |
| | **WML** | 71.52 | **65.20** | **69.32** | **71.44** | **72.76** | **73.10** |

different pruning ratios to match Classifier1-3 in self-distillation [61], and Model4 and Classifier4 are the same as the baseline network. More experiments with other pruning ratios for WML can be found in Sec.4.4. We adapt our framework with several ResNets [21] and also the MobileNetV2 [45], where results are present in Table 1 and Table 2, respectively.

There are two interesting pieces of evidence showing the advantage of the proposed WML on CIFAR100. **Firstly**, we can observe that WML leads to significant accuracy boosts, where our peer Model4 outperforms baseline networks with an increment of 2.29% to 3.56% on CIFAR100 for the three networks. In addition, our peer models beat the most classifiers in the three self-distillation baselines under similar model sizes, verifying the superiority of the proposed framework over self-distillation. Although SCAN achieves better results in Model1 and Model2 for ResNet18, we should notice that SCAN adds additional attention modules over SD, which has more parameters than SD and our WML. When the peer model is small, the additional attention module that contains relatively numerous parameters plays a more important role to improve the performance, while whose advantage decreases with the peer models become larger. **Secondly**, compared with self-distillation baselines, which usually encounter significant performance drops in the shallower classifiers, our peer models can maintain similar performance with moderate network pruning, especially on the large networks. Taking the Model2 on ResNet50 as an example, whose model size is only 50% as the baseline, our WML obtains 79.69% accuracy, while SD only achieved 74.21% accuracy. SCAN added additional attention modules to further improve self-distillation performance, achieving 78.34% accuracy while still worse than our WML. The above results demonstrate that reducing the width is a better way to get more competitive peers than the self-distillation paradigm using the shallow models. The comparison results on ImageNet in Table 2 again verify the advantage of the proposed framework, where our WML also outperforms the self-distillation baseline under different model sizes on the ImageNet dataset. As shown in the last column of Table 1, which is the ensemble of the four peer models, our WML outperforms the three baselines by large margins. This superiority comes from two aspects. One is that our weighted mutual learning can boost all peers' performance, thus enhancing the ensemble. The other is that our optimized weights can capture the peers' importance more precisely.

## 4.2 Comparison with Online Distillation

Table 3 compares the proposed WML with several online distillation methods, including RKD[41], CTSL-MKT[41], DML [70], ONE [71], and self-distillation (SD) [61] on CIFAR10, CIFAR100, and ImageNet. In these online distillation methods, we consider four peer models as our WML for fair comparisons, and all reported results for baselines can also be found in [63]. We can see that knowledge distillation and online distillation both outperform the directly trained baseline model with a large margin, showing the effectiveness of knowledge distillation. More interesting, different from traditional knowledge distillation which requires a powerful and large teacher model, online distillation methods can learn more meaningful information from peer models and gain performance improvements without a teacher. These comparison results again verified the effectiveness of the proposed framework, which outperforms most online distillation methods on different datasets and neural networks.

## 4.3 Comparison with Channel Pruning

In our WML, we leverage channel pruning to generate peer models with different pruning ratios at initialization. As a byproduct, we can get a series of pruned models with different FLOPs as peer models. In this subsection, we compared the performance of the pruned models generated by our WML with existing channel pruning methods. For fair comparisons, we conduct experiments with ResNet32 and ResNet56 on CIFAR10, and MobileNetV2 on ImageNet, which are the common setting to compare with existing works [63]. One clear advantage of WML over existing channel

Table 3: Top-1 accuracy (%) comparison results with online-distillation.

| Datasets | Models | Baseline | KD[25] | RKD[41] | MKT[41] | DML[70] | ONE[71] | SD[61] | **WML** |
|---|---|---|---|---|---|---|---|---|---|
| CIFAR10 | ResNet18 | 94.25 | 94.67 | 94.98 | 95.33 | 95.19 | 95.56 | 95.87 | **95.97** |
| | ResNet50 | 94.69 | 94.56 | 95.23 | 95.76 | 95.73 | 95.85 | 96.01 | **96.17** |
| CIFAR100 | ResNet18 | 77.09 | 77.79 | 76.43 | 77.46 | 77.54 | 77.87 | 78.64 | **79.38** |
| | ResNet50 | 77.42 | 79.33 | 78.02 | 78.52 | 78.31 | 78.52 | 80.56 | **81.24** |
| ImageNet | MobileNetV2 | 71.52 | 72.23 | - | 72.46 | 72.29 | 72.20 | 72.37 | **72.76** |

Table 4: Comparison with channel pruning methods for ResNet-32 and ResNet-56 on CIFAR10

| Models | Method | Baseline Acc | Pruned Acc | Acc Drop↓ | FLOPs Drop↓ |
|---|---|---|---|---|---|
| ResNet32 | LCCL [8] | 92.33% | 90.74% | 1.59% | 31.2% |
| | SFP [22] | 92.63% | 92.08% | 0.55% | 41.5% |
| | GFP40 [38] | 93.51% | 92.86% | 0.65% | 40.1% |
| | GFP50 [38] | 93.51% | 92.64% | 0.87% | 50.1% |
| | PScratch [52] | 93.18% | 92.18% | 1.00% | 50.0% |
| | FPGM [24] | 92.63% | 92.31% | 0.32% | 41.5% |
| | WML30 | 92.63% | 92.99% | **-0.36%** | 24.4% |
| | WML40 | 92.63% | 92.23% | **0.40%** | 45.4% |
| | WML50 | 92.63% | 92.10% | **0.53%** | 50.0% |
| ResNet56 | PFEC [31] | 93.04% | 93.06% | -0.02% | 27.6% |
| | LCCL [8] | 94.35% | 92.81% | 1.54% | 37.9% |
| | HRank [35] | 93.26% | 93.52% | -0.26% | 29.3% |
| | NISP [58] | 93.26% | 93.01% | 0.25% | 35.5% |
| | AMC [23] | 92.80% | 91.90% | 0.90% | 50.0% |
| | GFP40 [38] | 93.68% | 93.54% | 0.14% | 40.2% |
| | GAL [36] | 93.26% | 93.38 % | -0.12% | 37.6% |
| | WML30 | 93.26% | 93.93% | **-0.67%** | 24.7% |
| | WML40 | 93.26% | 93.46% | **-0.20%** | 41.7% |
| | WML50 | 93.26% | 92.68% | **0.58%** | 49.0% |

pruning methods is that we can get a series of pruned models with different FLOPs in a single run. Table 4 and 5 compared our pruned peer models with existing baselines on CIFAR10 and ImageNet. We make the peer models toward $\{30\%, 40\%, 50\%\}$ pruning ratios on FLOPs for ResNet32 and ResNet56 on CIFAR10, and $\{30\%, 50\%, 70\%\}$ for MobileNetV2 on ImageNet.

As shown in Table 4, there are two interesting findings. **First**, WML obtains higher accuracy than all channel pruning competitors on CIFAR10 after pruning similar FLOPs. For example, the model with 40% pruned FLOPs by WML only has a 0.40% accuracy drop for ResNet32 and even 0.20% performance increase for ResNet56, which both outperform the competitors on CIFAR10. **Second**, our pruned models with moderate pruning achieve better performance than the baseline model while with much fewer FLOPs. This benefit comes from the online distillation as we find the SNIP baseline in [40], which adopts the same pruning method as us while without knowledge distillation, obtains worse results than those elaborately designed pruning methods. However, combined with online distillation, our pruned model can achieve better performance than elaborately pruned models. A straightforward direction for future work is to apply those competitive pruning methods into our framework for further performance improvements.

Table 5 presents the comparison results of our pruned peer models on ImageNet, which also achieves competitive results compared with other SOTA channel pruning methods under different FLOPs constraints. For example, when the FLOPs are reduced by about 30%, WML30 only has 0.08% Top1 accuracy drop, which is better than MetaPruning's 0.60%, AMC's 1.00%, and DMCP's 0.10% at the similar pruning ratio. A similar advantage can also be found when pruning 50% and 70% FLOPs.

Table 5: Comparison with channel pruning methods for MobileNetV2 on ImageNet

| Models | Method | Baseline Acc | Pruned Acc | Top1 Drop↓ | FLOPs Drop↓ |
|---|---|---|---|---|---|
| MobileNetV2 | MetaPruning30 [39] | 71.80% | 71.20% | 0.60% | 27.7% |
| | MetaPruning50 [39] | 71.80% | 68.20% | 3.60% | 53.3% |
| | MetaPruning70 [39] | 71.80% | 63.80% | 8.20% | 71.0% |
| | AMC [23] | 71.80% | 70.80% | 1.00% | 29.7% |
| | DMCP [17] | 72.30% | 72.20% | 0.10% | 29.7% |
| | Group [38] | 75.74% | 69.16% | 6.58% | 50.0% |
| | WML30 | 71.52% | 71.44% | **0.08%** | 29.8% |
| | WML50 | 71.52% | 69.32% | **2.20%** | 52.9% |
| | WML70 | 71.52% | 65.20% | **6.32%** | 63.1% |

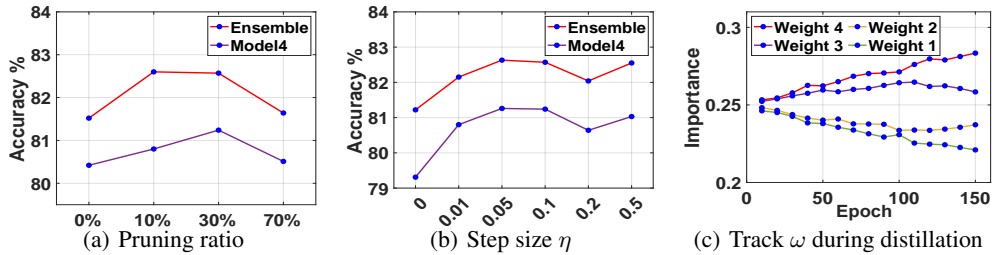

Figure 2: Ablation studies on pruning ratio, step size $\eta$ and peer importance.

## 4.4 Ablation Studies

In this section, we first conduct ablation studies to investigate whether the dynamic weighting strategy and peer pruning help improve the performance of online distillation. Then, we show increasing the number of peers gradually enhance the performance of the proposed framework.

Figure 2 (a) presents how the performance of Model4 and ensemble change with different pruning levels. As shown, when slightly pruning peers (only prune less than 10% parameters for Model1-3), WML can get better results than the unpruned framework in terms of both Model4 and ensemble performance. An underline reason is that slightly pruning rarely decreases the performance of peer models, while the diversity it brings can further enhance the distillation. However, when the pruning ratio is too high (the last data point in the Figure 2 (a) prunes more than 70% parameters for Model1-3),we can see a clear performance drop since the pruning damages the performance of Model1-3 seriously, which also further affects the distillation for Model4. Figure 2 (b) presents how the dynamic weighting strategy works in our WML, where the pruning ratio is set as Sec.4.1. As shown, the first data point in Figure 2 (b) indicates that we follow deep mutual learning [70] that remove the $\omega$ in Eq.(1), and the remaining points are with different $\eta$ to update $\omega$. We can see a clear performance boost from unweighted distillation to weighted distillation, which verifies the effectiveness of our dynamic weighting strategy, which is also robust to the step size $\eta$. In Figure 2 (c), we prune Model 1-3 with 75%, 55%, and 20% parameters to match with Classifier 1-3 in [61], and the four lines in Figure 2 (c) track the change of weights $\omega$ for each peer model. As shown, the weight of Model4 and Model4 consistently increase since they contains much more parameters and with better generalization ability. In contrast, the weights for the other two models both decrease, and Model1 with the least parameters descents the fastest. This phenomenon also verifies our weighting strategy can adaptively adjust the importance of peers in distillation.

The main experiments show good results of the proposed framework with four peers. We also verify the superiority of WML with different number of peers in Figure 3. For $M = 2$, we leave one peer unpruned and the other is pruned with 50%, and for $M = 4, 8$, we leave one peer unpruned and the remained peer models are pruned gradually from 30% to 70%. As show in Figure 3, WML scale well with more peers although gain decrease with the number of peers increase. The phenomenon verifies that more knowledge from peers can further enhance the performance of the students.

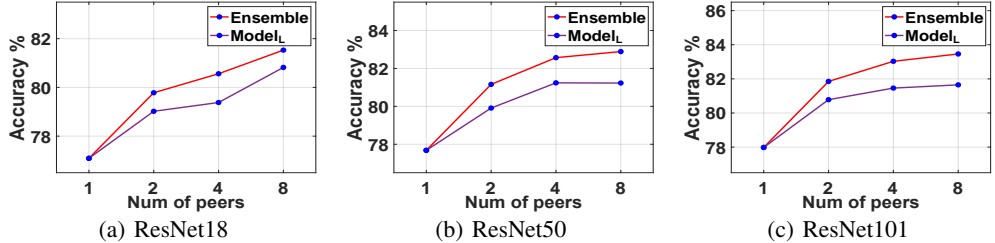

Figure 3: Performance with the number of peers under three different networks.

## 5 Conclusion and Future Work

This paper proposes a new framework for online distillation, called Weighted Mutual Learning with Diversity-Driven Model Compression (**WML**), which introduces the structured network pruning to obtain diversified peer models, and a bi-level formulation is further leveraged to estimate the relative importance of peers in forming a powerful teacher during the online distillation. The WML innovatively combines network pruning and online distillation to enhance performance and compress models simultaneously. Extensive experimental results have verified the proposed framework, which leads to consistent accuracy improvements on various neural networks and datasets under different model sizes. This paper considers a simple and efficient training-free channel pruning method to get the peer models while which is not as competitive as recent training-based channel pruning methods. Combining more advanced channel pruning methods with online distillation for further performance and efficiency boost and extending WML to feature distillation or other applications, e.g., object detection, natural language processing, and architecture search, are among directions for future work.

## Acknowledgments and Disclosure of Funding

This work was partially supported by Independent Research Fund Denmark under agreements 8022-00246B and 8048-00038B, the VILLUM FONDEN under agreement 34328, and the Innovation Fund Denmark centre, DIREC. This work was also in collaboration with Digital Research Centre Denmark – DIREC, supported by Innovation Fund Denmark.

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
