# OpenReview forum: "Weighted Mutual Learning with Diversity-Driven Model Compression"
_NeurIPS.cc/2022/Conference — NeurIPS 2022 Accept_

### Official Review · Reviewer_MvTh · 2022-07-11

**Rating:** 5
**Confidence:** 4
**Soundness:** 3 good
**Presentation:** 3 good
**Contribution:** 2 fair

**Summary:**

This paper introduces the structured network pruning and bi-level formulation to online distillation and shows its effectiveness in performance and efficiency. On the one hand, this paper proposes to prune unimportant channels in peer models, reducing the memory requirements. On the other hand, this method leverages a new formulation to estimate the relative importance of peer models, and improve the effectiveness of online distillation. Experimental results show that the proposed method outperforms a variety of methods in knowledge distillation and channel pruning.

**Questions:**

- How does the proposed method compare with the newly proposed methods in knowledge distillation and channel pruning?

- How does channel pruning contributes to the diversity of peer models?

- Why choose SNIP as the pruning method? Does it perform much better than others?

**Limitations:**

Yes

**Strengths And Weaknesses:**

Pros:

- The paper is very clearly written and easy to understand.

- The proposed method reduces the memory cost of online knowledge distillation and maintains competitive performance on a variety of deep neural networks.

- This paper makes a comprehensive ablation study on the proposed method.

-----------
Cons:

- This paper proposes to have improved the diversity of peer models but does not analyze how the proposed method impacts the diversity of all peer models deeply. It is necessary to discuss it in the experiment section.

- To my understanding, SNIP is an unstructured pruning method that prunes out the unimportant connections. I think the authors should consider other competitive structured pruning methods rather than just borrowing SNIP for structured pruning.

- Most of the selected competitors have been proposed for a long time and have a better alternative. The superiority presented in the experiments is not very convincing.

---

> ### Author Response · Authors · 2022-08-02
> **Response**
>
> Thanks for the constructive comments! We update our Appendix and also release our codes. We hope our answers below address all your concerns.
>
> 1. How does channel pruning contributes to the diversity of peer models?
>
> We conducted ablation studies in Sec 4.4 to show the effectiveness of diversity enhancement, and **more ablation studies are also added in Appendix D**. As we know, the model generally decreases its generalization ability with the pruning ratio increase. However, we find that, in our WML, when the pruning ratio is mild, our models can be trained even better than the unpruned scenarios, e.g., as shown in Figure 2 (a) and Figure 7 (a). In addition, we further compare our WML without our dynamic weighting strategy (WML-P) with collaborative learning (CL), in which our WML-P only prune peers in CL under different ratios. The results in Table 6 show that our WML-P can achieve even better results than CL, verifying the effectiveness of pruning enhanced diversity.
>
> 2.How does the proposed method compare with the newly proposed methods in knowledge distillation and channel pruning?
>
> Thanks for the constructive suggestion, and we reorganize Sec 4.2 and 4.3. We **add two more recent online distillation, CTSL-MKT, RKD, in Table 3**. In addition, we add a new section in Appendix E to compare with more existing knowledge distillation and self-distillation, in which **we add 7 more recent baselines for comparison**. In addition, we also **add two recent channel pruning baselines [1,2] in Table 4 for better comparison**.
>
> 3.Why choose SNIP as the pruning method? Does it perform much better than others?
>
> We consider SNIP as it is extremely efficient and also competitive compared with conventional pruning techniques. For example, although a SOTA channel pruning method GFP [1] obtains better accuracy compared with SNIP baseline, it requires much higher computation cost as it needs to finetune the pruned model after pruning each channel. In addition, by incorporating our weighted mutual learning, we can achieve better accuracy than GFP under the SNIP pruning strategy. Apart from SNIP, we also introduce another two training-free pruning methods, GraSP and SynFlow, into our WML. Table 7 in Appendix compares the three training-free pruning strategies in our WML. In general, SynFlow achieves similar results as SNIP with marginal performance drop in several cases, while GraSP obtains worse results. This phenomenon is also in line with existing pruning at initialization research.
>
> [1] Group fisher pruning for practical network compression, ICML2021
>
> [2] Pruning from scratch. AAAI, 2020

---

### Official Review · Reviewer_WYtc · 2022-07-11

**Rating:** 5
**Confidence:** 3
**Soundness:** 2 fair
**Presentation:** 2 fair
**Contribution:** 2 fair

**Summary:**

This paper tackle the problem of online distillation using weighted mutual learning, where structured network pruning is leveraged to reduce the memory consumption. Extensive experiments on several neural networks and three datasets show this method outperforms existing online distillation and self-distillation methods.

**Questions:**

Q1: There is nothing in the supplementary material. Did the authors submit a wrong file?

Typo:

Table 1 ResNet 101->WML->Model 1: "79.6" -> "79.60".

**Limitations:**

See weaknesses.

**Strengths And Weaknesses:**

Strengths:
1. The idea is interesting and can be applied in many tasks.
2. Experimental setups are proper and there are sufficient experiments.


Weaknesses:
1. The motivation of the method is not described clearly. For example, why do you propose to use several peer models to share different parts in a hierarchical structure in Figure 1? If the motivation does not described clearly, the method seems to be incremental.
2. The improvements are marginal. In Table 1,2,3, most of the accuracy gaps between WML and SCAN/SD are within 1%, which is marginal. Did the authors run the experiments for several different times and take the means and standard variances?
3. Tables need to be improved. More detailed descriptions should be added in the table caption. For example, the values in Table 1 and Table 3 are accuracy (%) right? The authors can clearly mention that.

---

> ### Author Response · Authors · 2022-08-02
> **Response**
>
> Thanks for the constructive comments! We update our Appendix and also release our codes. We hope our answers below address all your concerns.
>
> 1. The motivation of the method is not described clearly.
>
> Thanks for the constructive comments. The main motivation of this paper of using hierarchical structure is to reduce the memory consumption and enhance the diversity among peers in online distillation simultaneously. First, We first adopt a hierarchical structure to reduce the memory consumption compared with online distillation as peers share parts. Second, since peers share different parts, diversity among peers can be maintained. Compared with online distillation (peers are independent ) and auxiliary head based distillation (peers shares the same part), the hierarchical structure can obtains a good trad-off between diversity and memory consumption. More important, with the hierarchical structure, we are able to prune the peers under different pruning ratios to further reduce the memory consumption and also enhance the diversity among peers.
>
> The motivation of our dynamic weighting strategy is that the peers are with different generalization abilities, and appropriately accounting for the importance of peers can enhance the performance of the ensemble to provide better supervision. We revise accordingly to highlight our contribution for more clarification in the introduction.
>
> 2. The improvements are marginal, most of the accuracy gaps between WML and SCAN/SD are within 1%, which is marginal. Did the authors run the experiments for several different times and take the means and standard variances?
>
> In the Table1, we compared the peer models with different model sizes under different knowledge distillation strategies. We can see that, our ensemble models generally outperform the three self-distillation baselines from 0.89% to 1.90% on CIFAR100. Another important aim of knowledge distillation is to get a compact model with competitive performance, to deploy in the computation-limited environment, e.g., edge devices, and we are more interested in the small models. Taking the Model 1 in Table 1 (with highest pruning ratio) as example, our WML obtains more than 5\% accuracy improvement than baselines in ResNet50, and more than 7\% accuracy in ResNet101.
>
> Although our superiority gradually becomes marginal with the model size increase (we only report the performance for the largest peer in Table2, 3), we should notice that this is due to the large models with excellent generalization ability, and it is not easy to obtain significant improvement on the basis of large models under different training schemes.
>
> Following most self-distillation papers, we evaluate each model for 3 times to report the average.
>
> 3. Tables need to be improved. More detailed descriptions should be added to the table caption. For example, the values in Table 1 and Table 3 are accuracy (%) right? The authors can clearly mention that.
>
> Thanks for the constructive comments. We have revised the table caption accordingly for clarification.
>
> 4. Wrong file in the supplementary material.
>
> Thanks for the comments. We edit the new supplementary material and also release our codes.

---

> ### Author Response · Authors · 2022-08-09
> **We would love to hear your feedback on our rebuttal**
>
> Dear Reviewer WYtc,
>
> As the discussion period is close to the end and we have not yet heard back from you, we wanted to reach out to see if our rebuttal response has addressed your concerns.
>
> We are more than happy to discuss further if you have any further concerns and issues, please kindly let us know your feedback. Thank you for your time and help!

---

> > ### Comment · Reviewer_WYtc · 2022-08-09
> > **Update**
> >
> > Thanks to the authors for the rebuttal. I think my concerns have been addressed. I have raised my rating.

---

### Official Review · Reviewer_1K3n · 2022-07-11

**Rating:** 6
**Confidence:** 4
**Soundness:** 3 good
**Presentation:** 3 good
**Contribution:** 2 fair

**Summary:**

The paper proposed an online-distillation framework called Weighted Mutual Learning with Diversity-Driven Model Compression (WML). The authors Introduce a hierarchical structure in which peer networks share different parts to reduce memory consumption. A structured pruning method based on SNIP is adopted for the diversity of peer students. They use a dynamic weighting method to determine the importance of the pruned peers and formulate it as a bi-level problem.

**Questions:**

--- The contributions are the memory-friendliness and diversity of student peers; they respectively benefit from a hierarchical structure where peers share different parts and structure pruning. As mentioned in the original paper, the hierarchical architecture is adopted from [1] and the pruning criterion is SNIP[2]. Neither of these points is original, and the innovation of this paper needs more clarification.

-- Though self-distillation can be regarded as a special case of online distillation, it is different from the mutual learning method in this paper since it only uses its own structure as a teacher. Instead of self-distillation, the comparison with online distillation using multiple peers should be of more concern. In this paper, only DML[3] and ONE[4] are compared without more updated methods. This makes a claim in line 66 "significantly outperforms existing online distillation and self-distillation methods" less convincing. I think Table 3 should be expanded and highlighted with more comparisons with the latest online distillation approaches.

-- The authors mention the use of structured pruning to enhance diversity, but due to the use of a partially shared hierarchical structure, diversity is limited compared to other online methods that can use completely different networks. It seems to me that the advantage of structural diversity here is only in comparison with self-distillation methods or [1].

-- The diagram in Figure 1 (b) is ok, but it could easily be confused with self-distillation. It would be more representative of online distillation if it were changed to different peer networks.

[1] G. Song and W. Chai. Collaborative learning for deep neural networks. NeurIPS, 2018.
[2] N. Lee, T. Ajanthan, and P. Torr. Snip: Single-shot network pruning based on connection sensitivity. ICLR, 2019.
[3] Y. Zhang, T. Xiang, T. M. Hospedales, and H. Lu. Deep mutual learning. CVPR, 2018.
[4] X. Zhu, S. Gong, et al. Knowledge distillation by on-the-fly native ensemble. NeurIPS, 2018.

**Ethics Review Area:**

["I don’t know"]

**Limitations:**

Yes，it's oK.

**Strengths And Weaknesses:**

-- The paper is clearly presented, and the main ideas are easy to follow.

-- The two difficulties of online distillation: memory consumption and model diversity, as the proposed framework tries to solve, are important and meaningful. The authors conduct several experiments to prove the effectiveness.

-- Structured network pruning is used to generate peer networks in the hierarchical structure. This combines pruning and online distillation. And to dynamically decide the weight of each peer rather than take the average, the authors formulate it as a bi-level optimization problem.

---

> ### Author Response · Authors · 2022-08-02
> **Response**
>
> Thanks for the constructive comments! We update our Appendix and also release our codes. We hope our answers below address all your concerns.
>
> 1. The pruning criterion and hierarchical structure are not novel, and the innovation of this paper needs more clarification.
>
> In this paper, we combine the hierarchical structure and structure pruning to reduce memory consumption, which is a novel trial for online distillation. We first adopt a hierarchical structure to reduce memory consumption while maintaining the diversity among peers in online distillation since the peers share different parts. On the other hand, we prune the peers under different pruning ratios to further reduce memory consumption and further enhance the diversity among peers. Another main contribution of this paper is to devise a closed-form gradient to optimize the weights, and to account for the importance of peers appropriately. **We revise accordingly to highlight our contribution for more clarification in the Introduction for more clarification.**
>
> 2. I think Table 3 should be expanded and highlighted with more comparisons with the latest online distillation approaches.
>
> Thanks for the constructive suggestion, and we rephrase this expression in Sec. 4.2 accordingly. In addition, we reorganize Sec 4.2, and also **add two more online distillations, CTSL-MKT, RKD, in Table 3**. In addition, **we add a new section in Appendix E to compare with more existing knowledge distillation and self-distillation, in which we introduce 7 more baselines for comparison.**
>
> 3. Diversity is limited compared to other online methods. The effectiveness of diversity needs more analysis.
>
> Compared with traditional online learning, the hierarchical structure (or self-distillation) has less diversity freedom as each peer model contains fewer independent parts. However, the hierarchical structure has much less memory consumption than traditional online learning, and [1] also verify the hierarchical structure can even improve performance compared with online learning. In addition, our structure pruning can further enhance diversity among peers.
>
> We conducted ablation studies in Sec 4.4 to show the effectiveness of diversity enhancement, and **more ablation studies are also added in Appendix D**. As we know, the model generally decreases its generalization ability with the pruning ratio increase. However, we find that, in our WML, when the pruning ratio is mild, our models can be trained even better than the unpruned scenarios, e.g., as shown in Figure 2 (a) and Figure 7 (a). In addition, we further compare our WML without our dynamic weighting strategy (WML-P) with collaborative learning (CL) [1], in which our WML-P only prune peers in CL under different ratios. The results in Table 6 show that our WML-P can achieve even better results than CL, verifying the effectiveness of pruning enhanced diversity.
>
> 4. The diagram in Figure 1 (b) is ok, but it could easily be confused with self-distillation. It would be more representative of online distillation if it were changed to different peer networks.
>
> Thanks for this suggestion. We have revised accordingly. In addition, we also draw another four representative knowledge distillation framework in Figure 4 of Appendix, including feature distillation,  On-the-fly Native Ensemble (ONE), Self-Distillation, and Collaborative Learning.
>
> [1] G. Song and W. Chai. Collaborative learning for deep neural networks. NeurIPS, 2018.

---

> > ### Comment · Reviewer_1K3n · 2022-08-09
> > **Update**
> >
> > Thank you for your responses！ The revised version address most of my concerns, and I would like to raise my rating.

---

### Official Review · Reviewer_GDJC · 2022-07-16

**Rating:** 7
**Confidence:** 4
**Soundness:** 3 good
**Presentation:** 3 good
**Contribution:** 3 good

**Summary:**

This paper presents a new online distillation technique enabled by a novel bi-level formulation that is based on a closed form expression. The bi-level formulation results in a weighting of the peers according to importance thus enabling optimized training of the entire network of peers unlike prior works that depended on approximations. The proposed method enables systematic pruning of the networks which results in significant improvement over the state of the art in terms of accuracy given a certain number of flops.

**Questions:**

Having a closed form expression is of course impressive and handy, but in most cases a Taylor approximation provides a powerful approximation. So what accounts for the great improvement over the state of the art?

**Limitations:**

The authors have pointed out that to maintain a bounded investigation they have refrained from incorporating conventional pruning techniques which they say they will adopt in future work to get even better results.
That is the one discussion of limitations that I could find.

There is no mention of social impact, positive or negative. I would suggest that the proposed fine-grained optimization will have only positive social impact because it will help reduce complexity in a systematic manner, thus more energy efficient and compact devices can be constructed which is a net positive for society.

**Strengths And Weaknesses:**

Strengths:
1. Originality and Significance: The paper's proposed closed form approach is a clear advancement of the state of the art and is thus both original and significant.
2. Quality: The proposed closed form approach provides a compact and systematically tunable framework that achieves much better than the state of the art performance. The quality is therefore high.
3. Clarity: The paper is very well written. It provides a broad overview and then covers the state of the art in an insightful way and motivates the proposed technique. The technique is itself well described and the results are presented clearly as are the conclusions. The clarity is high.

Weaknesses
1. Having a closed form expression is of course impressive and handy, but in most cases a Taylor approximation provides a powerful approximation. So what accounts for the great improvement over the state of the art?
2. How exactly the closed form expression is  being used needs to be explained better.
3. The above are minor weaknesses. The overall quality of the paper far outweighs these "weaknesses."

---

> ### Author Response · Authors · 2022-08-02
> **Response**
>
> Thanks for the constructive comments! We update the Appendix and also release our codes. We hope our answers below address all your concerns.
>
> 1. Taylor approximation provides a powerful approximation. So what accounts for the great improvement over the state of the art?
>
> The Taylor approximation is popular in many deep learning applications, while which also brings approximation errors during the training. To reduce errors, it is usually supposed to consider more terms in the finite-difference approximation, while which makes it less efficient. In contrast, rather than using the Taylor expansion approximation, we derive a closed-form when $\mathcal{L}_1$ has a specific structure, so as eliminate the approximation error and improve the efficiency.
>
> More importantly, the dynamic weighting strategy, which is based on the close-form derived gradient, enables the online distillation to adaptively determine the importance of each peer in distilling knowledge from a group of peers. And our ablation study in Figure 5 in Appendix C also verified the superiority of the dynamic weighting strategy, where our weighted ensemble achieved better generalization ability than the average ensemble. In addition, we leverage structured network pruning to improve the diversity among peers, so as further enhance the knowledge distillation process. Figure 2 in Sec 4.4 and Figure 5, 6 in Appendix D verified the superiority of this diversity-driven pruning.
>
> Overall, the performance improvement by our WML benefits from two parts, a hierarchy structure with pruning to enhance the diversity among peers, and a dynamic weighting strategy based on the close-form derived gradient to appropriately account the importance of peers.
>
> 2. How exactly the closed-form expression is being used needs to be explained better.
>
> Thanks for the constructive suggestion. We give more discussions in Section 3.3. We want to show that, when the loss function is with a specific structure, a weighted form, we can derive a close-form for the second-order information term $\frac{\partial^2 \mathcal{L}_1}{\partial \omega \partial \theta}$  for a large neural network. The closed-form derived gradient provides a theoretical foundation that a dynamic weighting strategy can appropriately account for the importance of peers.

---

### Author Response · Authors · 2022-08-08
**Paper updated and further discussion**

Dear reviewers,

We have uploaded a paper revision addressing your concerns and suggestions, and making other improvements:

1. Renewed Appendix and the released codes.

2. Added description on how exactly the closed-form expression is being used in Sec. 3.3.

3. Revised our contribution for more clarification in the Introduction.

4. Added two more online distillation baselines, CTSL-MKT, RKD, in Table 3. Added a new section in Appendix E to compare with more existing knowledge distillation and self-distillation, in which we introduce 7 more new baselines for comparison.

5. Added more ablation studies in Appendix D to show the effectiveness of diversity enhancement.

6. Revised table captions following suggestions.

We appreciate all reviewers for the hard work and helpful comments. We would like to address all reviewers’ concerns in the corresponding responses.

---

### Meta-Review · Area_Chair_uy56 · 2022-08-24

**Recommendation:** Accept
**Confidence:** Certain

**Metareview:**

This paper proposes an online distillation technique: Weighted Mutual Learning with Diversity-Driven Model Compression (WML). It uses a novel bi-level formulation that estimates the importance of the peers thus enabling optimized training of the entire network of peers. The proposed method got significant improvement over the state of the art in terms of accuracy. The proposed idea is novel and could be used in many tasks.



**Award:**

No

---

### Decision · Program_Chairs · 2022-09-14

Accept